# Switch to Generalize: Domain-Switch Learning for Cross-Domain Few-Shot Classification

**Zhengdong Hu**[1,2*]**, Yifan Sun**[2]**, Yi Yang**[3] [†]

[1] State Key Laboratory of Industrial Control Technology, Zhejiang University
[2] Baidu Research, China
[3] CCAI, College of Computer Science and Technology, Zhejiang University
{zhengdonghu,yangyics}@zju.edu.cn, sunyf15@tsinghua.org.cn

## Abstract

This paper considers few-shot learning under the cross-domain scenario. The cross-domain setting imposes a critical challenge, *i.e.*, using very few (support) samples to generalize the already-learned model to a novel domain. We hold a hypothesis, *i.e.*, if a deep model is capable to fast generalize itself to different domains (using very few samples) during training, it will maintain such domain generalization capacity for testing. It motivates us to propose a novel Domain-Switch Learning (DSL) framework. DSL embeds the cross-domain scenario into the training stage in a "fast switching" manner. Specifically, DSL uses a single domain for a training iteration and switches into another domain for the following iteration. During the switching, DSL enforces two constraints: 1) the deep model should not over-fit the domain in the current iteration and 2) the deep model should not forget the already-learned knowledge of other domains. These two constraints jointly promote fast generalization across different domains. Experimental results confirm that the cross-domain generalization capacity can be inherited from the training stage to the testing stage, validating our key hypothesis. Consequentially, DSL significantly improves cross-domain few-shot classification and sets up new state of the art.

## 1 Introduction

This paper challenges a realistic problem, *i.e.*, the cross-domain scenario, for few-shot learning. Basically, the few-shot learning task uses a classifier learned on the training set to recognize novel classes with very few support samples. In real-world applications, there is usually a domain gap between the training samples and the support samples Tseng et al. (2020); Guo et al. (2020); Chen et al. (2019). The domain gap further imposes a critical challenge: since the support samples are very rare, they do not suffice for mitigating the domain gap between the support set (for novel classes) and the training set. Consequentially, the domain gap significantly compromises the recognition accuracy of the novel classes. Therefore, learning a model with strong cross-modality generalization capacity is important for cross-domain few-shot learning Tseng et al. (2020).

We argue that we may enhance the desired cross-modality generalization by "learning-to-generalize". To be more specific, we hold a hypothesis / intuition that if a deep model learns to fast generalize itself to different domains (using very few samples) during training, it will maintain the good domain generalization capacity for testing.

We model our intuition into a novel Domain-Switch Learning (DSL) framework, as illustrated in Fig. 1. DSL uses multiple ($M > 1$) domains to construct cross-domain training scenario in a "fast switching" manner. Instead of simply mixing all the domains to construct a mini-batch (Fig. 1 (a)), DSL includes only a single domain into every training iteration and switches to another domain for the following iteration (Fig. 1 (b)). Moreover, each iteration contains very few samples per category, so as to imitate the few-shot setting. Therefore, after every switch, the deep model crosses into a domain different from the former one, yielding the cross-domain few-shot scenario.

---

[*]Zhengdong Hu makes his part of work during internship in Baidu Research.
[†]Corresponding author.

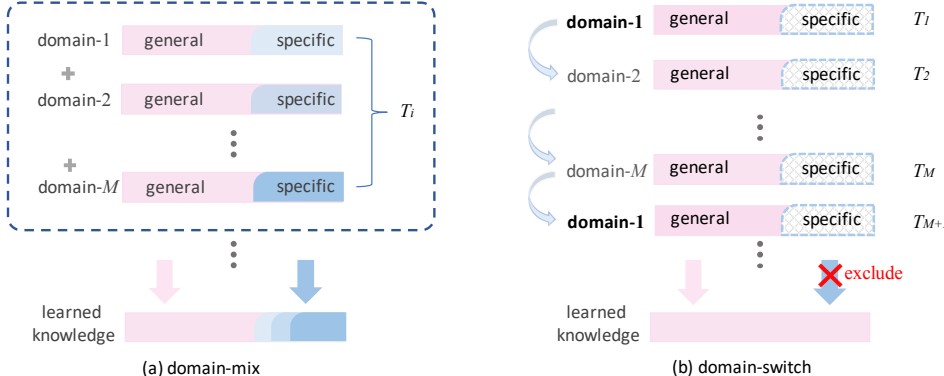

Figure 1: The comparison between Domain-Switch Learning (DSL) and a domain-mix pipeline. (a) mixes up all the $M$ domains in each training iteration. The deep model memorizes all the domain-specific knowledge (apart from the domain-general knowledge), which hinders cross-domain generalization. (b) DSL includes only a single domain into a training iteration and switches to another domain for the following iteration. Since the domain-specific knowledge does not fit other domains, the deep model is prone to discarding the domain-specific knowledge when switching to another domain. We note that in our experiments, we use multiple relatively small datasets ( *e.g.* CUB Welinder et al. (2010), Cars Krause et al. (2013), Places Zhou et al. (2018), and Plantae Horn et al. (2018) ) as the switchable domains while using mini-ImageNet as a basic training set which appears in each iteration.

An important advantage of this domain-switch learning manner is that it can suppress learning from domain-specific knowledge, as illustrated in Fig. 1. Specifically, we consider each domain contains both domain-general and domain-specific knowledge. If we mix all the domains for each training iteration (Fig. 1 (a)), the deep model may memorize all the domain-specific knowledge, as well as the domain-general knowledge. Consequentially, during testing, the domain-specific knowledge hinders generalization towards the novel domain and thus compromises few-shot classification accuracy. In contrast, in DSL (Fig. 1 (b)), the model only learns from one single domain in an iteration. Since the domain-specific knowledge of the current domain does not fit the following domain, the deep model is prone to discarding these domain-specific knowledge during the following iteration.

To further promote cross-domain generalization in this domain-switch learning scheme, DSL enforces two constraints as follows:

1) *The deep model should not over-fit the domain in the current iteration*, because over-fitting the current domain makes the model memorize much of the domain-specific knowledge. The first constraint is implemented with a domain-specific prompter module (consisted of multiple prompters). After the model learns from the $i$-th training domain $D_i$ and gets updated, its prediction accuracy on $D_i$ increases and is relatively high. We store this edition of model as the prompter for $D_i$. Next time before the training domain switches to $D_i$ again, the model just gets updated from $D_{i-1}$ and becomes relatively inaccurate on $D_i$. Given the accurate prompter and the inaccurate learner (*w.r.t.* $D_i$), we average their prediction for supervision so that the penalty on the learner will be suppressed. The details are to be accessed in Section 3.3.

2) *The deep model should not forget the already-learned knowledge of former domains*, so that next time the model crosses into these domains again, it can directly re-use the corresponding knowledge for prediction. The second constraint is implemented with a domain-general teacher module. Specifically, we collect several historical models and average their parameters to get a mean model. During training, we use the softmax prediction of the mean model as the auxiliary supervising signals for the learner (apart from the ground truth label). The mean model serves as a teacher distilling the already-learned knowledge of former domains to the learner. Since this teacher has no obvious bias towards any single domain, we name it as a domain-general teacher. The details of the domain-general teacher module are to be accessed in Section 3.4.

These two constraints achieve complementary benefits for DSL, and jointly reinforce the cross-domain generalization. Extensive experiments under four cross-domain scenarios show that DSL consistently improves cross-domain few-shot learning and achieves performance on par with the state-of-the-art methods.

Our main contributions are summarized as follows:

• We propose a novel Domain-Switch Learning (DSL) framework for cross-domain few-shot learning. DSL uses multiple domains for training and switches the domain in consecutive training iterations. It provides a cross-domain learning scenario where the deep model learns to generalize across different domains.

• Under the DSL framework, we integrate two modules, *i.e.*, the domain-specific prompter and the domain-general teacher. These two modules achieve complementary benefits for DSL and jointly reinforce the cross-domain generalization.

• We conduct extensive experiments to validate the effectiveness of the proposed DSL. Experimental results confirm that the fast generalization capacity can be inherited from training to testing and thus improves cross-domain few-shot classification. On all the four popular benchmarks, DSL achieves performance on par with the state of the art.

## 2 RELATED WORKS

**Few-Shot Learning.** Few shot learning methods could be roughly categorized into two branches: meta-learning and fine-tuning methods. Some meta-learning Satorras & Estrach (2018); Mishra et al. (2018); Ren et al. (2018); Snell et al. (2017); Sung et al. (2018); Vinyals et al. (2016) methods map the few-shot samples into a non-linear embedding space and evaluate the similarity between support samples and the query samples. Recently, fine-tuning methods Chen et al. (2019); Tian et al. (2020); Afrasiyabi et al. (2020) propose to pre-train the model on the training set (typically in a classification learning approach), and fine-tune the classifier on the novel classes. The fine-tuning methods achieve competitive performance compared with the state-of-the-art meta-learning methods. This work considers the few-shot learning task under the cross-domain scenario and adopts the fine-tuning pipeline as the baseline.

**Domain Generalization and Adaptation** may be viewed as two closely related concepts. Domain adaptation uses abundant (unlabeled) samples to adapt the already-learned model from the source domain to the target domain Ganin & Lempitsky (2015); Tzeng et al. (2017); Long et al. (2017; 2015; 2016); Tzeng et al. (2014). In contrast, domain generalization aims to generalize the knowledge from multiple source domains to the target domain without learning any samples of the target domain during the training stage Balaji et al. (2018); Li et al. (2018). In cross-domain few-shot learning, there is a domain gap between the training set and the testing set. To promote the cross-domain generalization, Tseng et al. (2020) considers the challenge is more of a domain generalization problem than a domain adaptation problem. This consideration is reasonable because the support samples are very rare and do thus not suffice for a typical domain adaptation.

From the viewpoint of domain generalization, FWT Tseng et al. (2020) proposes to simulate the cross-domain scenario to improve the domain generalization capacity. It indeed inspires us of the multi-domain training strategy. In spite of sharing the multi-domain training strategy, DSL significantly differs from FWT: First, FWT uses changes the training domain epoch-by-epoch. It does not rely on fast switching for better generalization. In contrast, DSL is featured for the novel fast switching manner and finds that this manner promotes learning domain-general knowledge. Second, DSL further considers two constraints in multi-domain training strategy, *i.e.,* the deep model should not over-fit every single domain and keep recognition on learned knowledge shared with other domains. Finally, DSL achieves superior cross-domain effect than FWT (Section 4.2).

**Catastrophic forgetting.** Catastrophic forgetting plays an important role in the mechanism of DSL. When the deep model is trained sequentially on multiple tasks (domains), there is a tendency that the knowledge learned on former tasks (domains) is abruptly lost, yielding the so-called catastrophic forgetting Kirkpatrick et al. (2017); McCloskey & Cohen (1989); French (1999).

In DSL, when the deep model switches from a domain to another, it forgets much of the knowledge on the former domain, due to catastrophic forgetting. Forgetting the domain-specific knowledge is

beneficial, while forgetting the domain-general knowledge is harmful. In response, next time the deep model comes across the former domain again, we 1) supplement it with the corresponding knowledge with a domain-specific prompter to prevent over-fitting the domain-specific knowledge and 2) use a domain-general teacher to reinforce the common knowledge shared among all the former domains. In a word, in its mechanism, DSL actually utilizes catastrophic forgetting to improve cross-domain generalization.

# 3 METHODS

## 3.1 PRELIMINARIES

In the cross-domain few-shot learning task, there is a significant domain gap between the training set and the testing set. The testing set consists the support set and query set, which are sampled from the same novel categories (which are unseen in the training set). The support set provides very few labeled samples for recognizing the unlabeled samples in the query. Concretely, the support set contains $C$ classes and each class has $K$ samples, which is a $C$-way $K$-shot setup in few-shot learning.

Our work is based on a popular fine-tuning pipeline Chen et al. (2019); Tian et al. (2020); Afrasiyabi et al. (2020). The fine-tuning baseline uses the training set $\mathcal{D}$ to learn a deep model consisted of a feature encoder $F$ and a feature classifier $Y$. $F$ is parameterized with $\theta$ and $Y$ is parameterized with a weight matrix $\boldsymbol{W}$. During training, given a sample $\boldsymbol{x}$, the feature encoder first maps $\boldsymbol{x}$ into a feature $F(\boldsymbol{x};\theta) \in \mathcal{R}^d$. Then the feature classifier $Y$ predicts the softmax probability of $\boldsymbol{x}$ belonging to each training category as $P(i|\boldsymbol{x};\theta,\boldsymbol{W}) = \frac{\exp(\boldsymbol{w}_i^{\mathrm{T}} F(\boldsymbol{x};\theta))}{\sum_j \exp(\boldsymbol{w}_j^{\mathrm{T}} F(\boldsymbol{x};\theta))}$ ($\boldsymbol{w}_i$ is the weight vector for the $i$-th category). The training is supervised with a popular classification loss function, *i.e.*, the cross-entropy loss, which is formulated as:

$$\mathcal{L}_{CE}(x) = -\log(P(y|\boldsymbol{x};\theta,\boldsymbol{W})) = -\log \frac{\exp(\boldsymbol{w}_y^{\mathrm{T}} F(\boldsymbol{x};\theta))}{\sum_j \exp(\boldsymbol{w}_j^{\mathrm{T}} F(\boldsymbol{x};\theta))}, \tag{1}$$

where $y$ is the label for $\boldsymbol{x}$, $\boldsymbol{w}_y$ is the weight vector corresponding to the ground-truth category.

During the testing stage, we replace the feature classifier $Y$ with a new linear classifier $L$. $L$ is a $C$-way classifier, corresponding to the total number of classes in the support set. We freeze the parameters of the feature encoder $F$ and fine-tune the parameters of $L$ with samples from the support set. Finally, we use the feature encoder $F$ and the learned $L$ as the predictor for the query.

This paper focuses on the training stage, *i.e.*, learning the feature encoder $F$ and the classifier $Y$ on the training set.

## 3.2 OVERVIEW OF DOMAIN-SWITCH LEARNING

Domain-Switch learning (DSL) employs $M$ datasets to construct the training set, *i.e.*, $\mathcal{D} = \{D_1, D_2, ..., D_M\}$. $D_i$ is an individual domain, which may be selected from some popular few-shot datasets, *e.g.*, CUB Welinder et al. (2010), Cars Krause et al. (2013), Places Zhou et al. (2018), Plantae Horn et al. (2018). These switchable training sets are relatively small, and does not suffice for learning a discriminative feature encoder. Therefore, following Tseng et al. (2020), we employ a relatively large-scale dataset mini-ImageNet Deng et al. (2009) as a "basic" training set. During training, each mini-batch actually contains samples from the basic mini-ImageNet and a switchable training set $D_i$. We note that such implementation does NOT contradict our motivation of using a single domain for each training iteration, because we may view the mixture of mini-ImageNet and a $D_i$ as a single "compound" domain.

During training, DSL uses samples from a single switchable domain $D_i$ (and the mini-ImageNet) for each iteration, as illustrated in Fig. 2. Since mini-ImageNet appears for every iteration, we omit it in Fig. 2 for simplicity. The domain-switching process is formulated as:

$$B_j \leftarrow D_i \mid D_i \in \mathcal{D}; \ i = j - \lfloor \frac{j}{M} \rfloor \times M \tag{2}$$

where $B_j$ is the training batch at the $j$-th iteration, $D_i$ is the *i-th* domain, $\lfloor \cdot \rfloor$ is the rounding down operation. We define a enumeration of all the $M$ domains as a **switching round**.

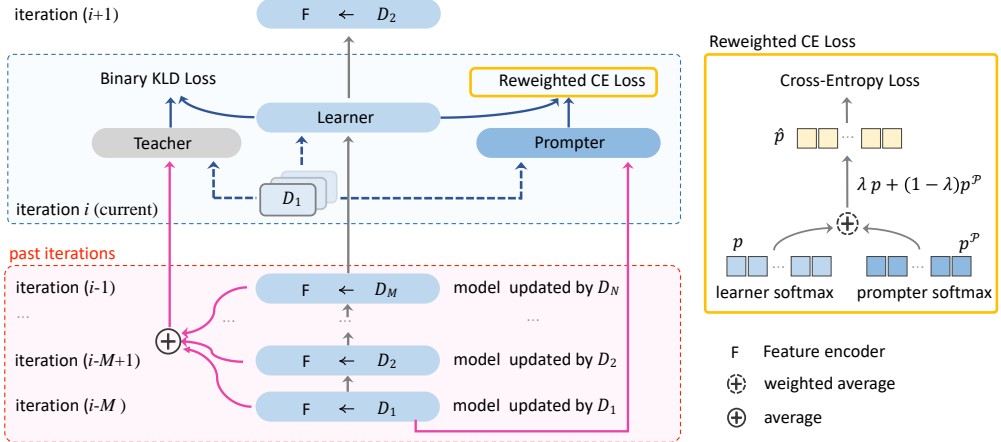

Figure 2: The overview of Domain-Switch Learning. DSL uses a single domain for a training iteration and switches to another domain for the following iteration. During the current iteration (the red area), DSL uses a domain-specific prompter and a domain-general teacher to assist the learner. The prompter prevents the learner from over-fitting the domain-specific knowledge of the current domain using a Re-weighted Cross-Entropy loss (Section 3.3). The teacher helps the learner to memorize the domain-general knowledge shared among former domains using a novel Binary Kullback-Leibler divergence loss (Section 3.4).

In order to further improve the cross-domain generalization in the above "fast switching", DSL enforces two constraints: 1) the deep model should not over-fit the domain in the current iteration and 2) the deep model should not forget the knowledge of other domains. These two constraints are implemented with a Domain-Specific Prompter (Section.3.3) and a Domain-General Teacher (Section.3.4) module, respectively.

## 3.3 DOMAIN-SPECIFIC PROMPTER

**Definition of the prompter**: Let us consider in the $j$-th iteration, the model has already learned from $D_i$ and gets updated. We store this edition of model as the prompter for the $(j + M)$-th iteration (when the learner is to learn from $D_i$ again), as illustrated in Fig. 2. The prompter aims to prevent the learner from over-fitting the current domain $D_i$.

Given a sample $\boldsymbol{x}$ in current domain, DSL averages the softmax prediction of the prompter ($P(y|\boldsymbol{x}; \boldsymbol{\theta}^{\mathcal{P}}, \boldsymbol{W}^{\mathcal{P}})$, where the upper-script $\mathcal{P}$ denotes the prompter) and the softmax prediction of the learner ($P(y|\boldsymbol{x}; \boldsymbol{\theta}, \boldsymbol{W})$), which is formulated as:

$$\hat{P} = \begin{cases} \lambda P(y|\boldsymbol{x}; \boldsymbol{\theta}, \boldsymbol{W}) + (1 - \lambda)P(y|\boldsymbol{x}; \boldsymbol{\theta}^{\mathcal{P}}, \boldsymbol{W}^{\mathcal{P}}), & \arg\max_i P(i|\boldsymbol{x}, \boldsymbol{\theta}^{\mathcal{P}}, \boldsymbol{W}^{\mathcal{P}}) = y \\ P(y|\boldsymbol{x}; \boldsymbol{\theta}, \boldsymbol{W}), & otherwise \end{cases} \tag{3}$$

where $\lambda$ is a hyper-parameter. The condition $\arg\max_i P(i|x; \boldsymbol{\theta}^{\mathcal{P}}, \boldsymbol{W}^{\mathcal{P}}) = y$ is to ensure the prompter makes the correct prediction.

With the re-weighted softmax prediction $\hat{P}$, the cross-entropy loss for supervising the classification is transformed into a Re-weighted Cross-Entropy loss (RCE loss in Fig. 2):

$$\mathcal{L}_{RCE} = -\log(\hat{\mathcal{P}}) \tag{4}$$

We explain how the domain-specific prompter prevents the learner from over-fitting the current domain with two following remarks (Remark1 and Remark2).

**Remark1**: Compared with the learner, the prompter has relatively higher predicted probability on the ground-truth category, *i.e.*, $P(y|\boldsymbol{x}; \boldsymbol{\theta}^{\mathcal{P}}, \boldsymbol{W}^{\mathcal{P}}) \geq P(y|\boldsymbol{x}; \boldsymbol{\theta}, \boldsymbol{W})$.

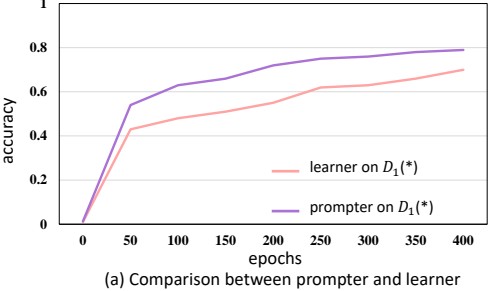 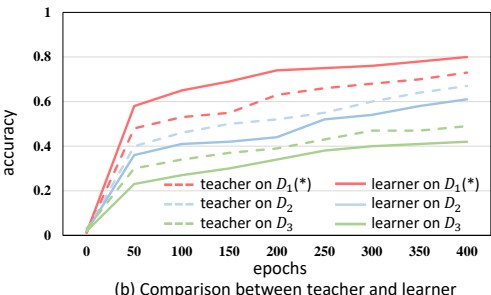

(a) Comparison between prompter and learner          (b) Comparison between teacher and learner

Figure 3: Analysis on the predictions of prompter and teacher. "$D_1(*)$" indicates that the current training domain is $D_1$. (a) The prompter fits $D_1$ better than the learner before the learner gets updated on $D_1$. (b) The teacher memorizes the general knowledge from all the domains and thus fits the former domains (*e.g.*, $D_2, D_3$) better, while the learner fits better $D_1$.

Remark1 is intuitive. Without losing generality, we assume the current learning domain in Fig. 2 is $D_1$. The domain-specific prompter is the former model updated by $D_1$ and thus fits $D_1$ well. In contrast, the current learner is newly updated by $D_M$ and forgets some knowledge of $D_1$, due to the well-known "catastrophic forgetting". Therefore, the prompter has higher predicted probability on the ground-truth category than the learner does. We empirically validate the above analysis in Fig. 3, from which we clearly observe that the prompter has higher prediction accuracy than the learner.

**Remark2**: Since $P(y|\boldsymbol{x}; \boldsymbol{\theta}^{\mathcal{P}}, \boldsymbol{W}^{\mathcal{P}}) \geq P(y|\boldsymbol{x}; \boldsymbol{\theta}, \boldsymbol{W})$ (*i.e.*, Remark1), the re-weighted procedure in Eq. 3 alleviates the penalty on the learner, which is beneficial to exclude domain-specific knowledge. Please refer to the Appendix for the conceptual proof. Consequentially, the prompter counters overfitting the current domain.

### 3.4 DOMAIN-GENERAL TEACHER

**Definition of the domain-general teacher**: We recall that a switching round is a period within which all the $M$ domains are enumerated. As illustrated in Fig. 2, we average the parameters of all the $M$ encoders updated in the past switching round to generate the domain-general teacher for the current domain $D_i$, which is formulated as:

$$\boldsymbol{\theta}^{\mathcal{T}} \leftarrow \frac{1}{M} \sum_{j=i-M}^{j=i-1} \boldsymbol{\theta}_j \tag{5}$$

The teacher aims to preserve and distill the domain-general knowledge learned in former domains to the current learner.

We note that the domain-general teacher is different from the popular *temporal moving average* for semi-supervised and unsupervised task Tarvainen & Valpola (2017); He et al. (2020); Grill et al. (2020). In temporal moving average, the teacher absorbs parameters from all the historical models (with higher weighting factor for the more recent model). In contrast, our domain-general teacher absorbs parameters from only $M$ historical models with equal weighting factor.

**Remark3**: When the learner gets updated from $D_i$, it has higher prediction accuracy on $D_i$ than the teacher does. Meanwhile, the teacher has higher prediction accuracy on former domains $D_j (j \neq i)$.

Remark3 is intuitive. It is because the teacher absorbs knowledge from $M$ historical learners, therefore preserving historical knowledge of the former domains. We empirically validate this intuition in Fig. 3 (b), which is consistent with Remark3. Moreover, since the teacher averages the parameters of all the $M$ historical learners, it has no obvious bias towards any single domain. Therefore, the teacher has relatively smaller variation of accuracy on all the domains, as shown in Fig. 3 (b).

**Binary Kullback-Leibler divergence loss.** Given a sample $\boldsymbol{x}$ in the current domain, we use the output of the teacher $P(y|\boldsymbol{x}; \boldsymbol{\theta}^{\mathcal{T}}, \boldsymbol{W}^{\mathcal{T}})$ as an auxiliary supervising signal for the learner. To this end, we propose a novel Binary Kullback-Leibler divergence (KLD) Loss as follows:

| Methods | | CUB | Cars | Places | Plantae |
|---|---|---|---|---|---|
| MatchingNet | s | $35.89 \pm 0.51$ | $30.77 \pm 0.47$ | $49.86 \pm 0.79$ | $32.70 \pm 0.60$ |
| RelationNet | s | $42.44 \pm 0.77$ | $29.11 \pm 0.60$ | $48.64 \pm 0.85$ | $33.17 \pm 0.64$ |
| MatchingNet+FT | s | $36.61 \pm 0.53$ | $29.82 \pm 0.44$ | $51.07 \pm 0.68$ | $34.48 \pm 0.50$ |
| RelationNet+FT | s | $44.07 \pm 0.77$ | $28.63 \pm 0.59$ | $50.68 \pm 0.87$ | $33.14 \pm 0.62$ |
| RelationNet + ATA | s | $43.02 \pm 0.40$ | $31.79 \pm 0.30$ | $51.16 \pm 0.50$ | $33.72 \pm 0.30$ |
| MatchingNet | m | $37.90 \pm 0.55$ | $28.96 \pm 0.45$ | $49.01 \pm 0.65$ | $33.21 \pm 0.51$ |
| RelationNet | m | $44.33 \pm 0.59$ | $29.53 \pm 0.45$ | $47.76 \pm 0.63$ | $33.76 \pm 0.52$ |
| MatchingNet + LFT | m | $43.29 \pm 0.59$ | $30.62 \pm 0.48$ | $52.51 \pm 0.67$ | $35.12 \pm 0.54$ |
| RelationNet + LFT | m | $48.38 \pm 0.63$ | $32.21 \pm 0.51$ | $50.74 \pm 0.66$ | $35.00 \pm 0.52$ |
| **DSL** | m | $\mathbf{50.15 \pm 0.80}$ | $\mathbf{37.13 \pm 0.69}$ | $\mathbf{53.16 \pm 0.88}$ | $\mathbf{41.17 \pm 0.80}$ |

Table 1: Comparison with the state of the arts for 5-way 1-shot task. "FT" ("LFT") denotes method in Tseng et al. (2020) , "ATA" denotes method in Wang & Deng (2021), "s"("m") denotes single-domain (multi-domain) learning.

| Methods | | CUB | Cars | Places | Plantae |
|---|---|---|---|---|---|
| MatchingNet | s | $51.37 \pm 0.77$ | $38.99 \pm 0.64$ | $63.16 \pm 0.77$ | $46.53 \pm 0.68$ |
| RelationNet | s | $57.77 \pm 0.69$ | $37.33 \pm 0.68$ | $63.32 \pm 0.76$ | $44.00 \pm 0.60$ |
| MatchingNet+FT | s | $55.23 \pm 0.83$ | $41.24 \pm 0.65$ | $64.55 \pm 0.75$ | $41.69 \pm 0.63$ |
| RelationNet+FT | s | $59.46 \pm 0.71$ | $39.91 \pm 0.69$ | $66.28 \pm 0.72$ | $45.08 \pm 0.59$ |
| RelationNet + ATA | s | $59.36 \pm 0.40$ | $42.95 \pm 0.40$ | $66.90 \pm 0.40$ | $45.32 \pm 0.30$ |
| NSAE | s | $68.51 \pm 0.76$ | $54.91 \pm 0.74$ | $71.02 \pm 0.72$ | $59.55 \pm 0.74$ |
| MatchingNet | m | $51.92 \pm 0.80$ | $39.87 \pm 0.51$ | $61.82 \pm 0.57$ | $47.29 \pm 0.51$ |
| RelationNet | m | $62.13 \pm 0.74$ | $40.64 \pm 0.54$ | $64.34 \pm 0.57$ | $46.29 \pm 0.56$ |
| MatchingNet + LFT | m | $61.41 \pm 0.57$ | $43.08 \pm 0.55$ | $64.99 \pm 0.59$ | $48.32 \pm 0.57$ |
| RelationNet + LFT | m | $64.99 \pm 0.54$ | $43.44 \pm 0.59$ | $67.35 \pm 0.54$ | $50.39 \pm 0.52$ |
| **DSL** | m | $\mathbf{73.57 \pm 0.65}$ | $\mathbf{58.53 \pm 0.73}$ | $\mathbf{74.10 \pm 0.72}$ | $\mathbf{62.10 \pm 0.75}$ |

Table 2: Comparison with the state of the arts for 5-way 5-shot task. "FT" ("LFT") denotes method in Tseng et al. (2020) , "ATA" denotes method in Wang & Deng (2021), "s"("m") denotes single-domain (multi-domain) learning.

$$\mathcal{L}_{BKLD} = P_y^{\mathcal{T}} \cdot log \frac{P_y^{\mathcal{T}}}{P_y} + (1 - P_y^{\mathcal{T}}) \cdot log \frac{1 - P_y^{\mathcal{T}}}{1 - P_y}, \tag{6}$$

where $P_y = P(y|\boldsymbol{x}; \boldsymbol{\theta}, \boldsymbol{W})$ and $P_y^{\mathcal{T}} = P(y|\boldsymbol{x}; \boldsymbol{\theta}^{\mathcal{T}}, \boldsymbol{W}^{\mathcal{T}})$.

The difference between the proposed BKLD loss and the popular KLD loss is: KLD aims to make the student consistent with the teacher at all the entries of the softmax prediction. In contrast, the proposed BKLD loss only focuses on the consistency at the ground-truth entry. In the experiment in Section 4.4, we empirically show BKLD loss is better than the KLD loss for DSL.

**Overall Training.** We recall that besides the switchable domains, mini-ImageNet is employed as a "basic" training set, which is sampled into all the training iterations. For mini-ImageNet, we adopt the cross-entropy loss as a basic loss $\mathcal{L}_{Basic}$. The overall training loss is as follows:

$$\mathcal{L} = \mathcal{L}_{Basic} + \alpha \cdot \mathcal{L}_{RCE} + (1 - \alpha) \cdot \mathcal{L}_{BKLD} \tag{7}$$

where $\alpha$ is a weighting factor.

## 4 EXPERIMENTS

### 4.1 EXPERIMENTAL SETUPS

We evaluate the proposed DSL on five datasets: *i.e.,* 4 fine-grained datasets (CUB Welinder et al. (2010), Cars Krause et al. (2013), Places Zhou et al. (2018), and Plantae Horn et al. (2018)) and 1 popular large scale dataset mini-ImageNet Deng et al. (2009). We use ResNet-10 He et al. (2016) without pretraining as the backbone network. Please refer to the Appendix for more experimental setups and implementation details.

| Methods | CUB | Cars |
|---|---|---|
| mini-ImageNet (single) | $63.76 \pm 0.60\%$ | $51.21 \pm 0.40\%$ |
| domain-mix (multi) | $64.73 \pm 0.68\%$ | $51.78 \pm 0.60\%$ |
| domain-switch (multi) | $\mathbf{66.33 \pm 0.71\%}$ | $\mathbf{53.61 \pm 0.68\%}$ |

Table 3: Comparison between two multi-domain learning scheme for 5-way 5-shot task.

## 4.2 COMPARISON WITH THE STATE OF THE ARTS

We compare DSL with the existing state of the arts on 4 benchmarks. We adopt the leave-one-out setting, *i.e.*, one of the fine-grained datasets is chosen as a testing set and the other 3 datasets (out of the 4 datasets) are used as the training sets. Meanwhile, we employ mini-ImageNet as a "basic" training set, which is available during the whole training stage. The results of 5-way 1-shot and 5-way 5-shot are summarized in Table 1 and Table 2, respectively.

From Table 1, we clearly observe the superiority of DSL under 5-way 1-shot classification. First, comparing DSL with all the single-domain training methods, we find that DSL achieves significant superiority. For example, DSL outperforms the most competing single-domain method ("Relation-Net + ATA (s)") Wang & Deng (2021) by +7.13%, +5.34%, +2%, +7.45% on CUB, Cars, Places and Plantae, respectively. Second, while the multi-domain training methods achieve considerable improvement (over their single-domain counterpart), DSL still outperforms them with a clear margin. For example, DSL outperforms the most competing multi-domain method ("RelationNet + LFT (m)") by +1.77%, +4.92%, +2.42%, +6.17% on CUB, Cars, Places and Plantae, respectively.

From Table 2, we clearly observe that DSL achieves competitive accuracy under the 5-way 5-shot classification. Most of the competing methods already appear in the comparison of 5-way 1-shot setting (Table 1). Compared with all these methods (except NSAE Liang et al. (2021)), DSL maintains significant superiority on all the four benchmarks. For example, DSL surpasses "RelationNet + LFT(m)"Tseng et al. (2020) by +8.58%, +15.09%, +6.75%, +11.71% on CUB, Cars, Places and Plantae, respectively.

NSAE in Table 2 is a very recent and competitive single-domain training method. It splits the support samples of the same category into two sub-sets for similarity learning. Therefore, it requires more than 1 support samples for each category and is incompatible to the 1-shot setting. With this unique similarity-learning manner, it achieves very competitive performance (*e.g.*, 68.51% on CUB). Compared with NSAE, DSL obtains consistent superiority For example, DSL surpasses NSAE by 5.06% on CUB.

## 4.3 THE EFFECTIVENESS OF THE DOMAIN-SWITCH OPERATION

A key characteristic of DSL is the domain-switch operation. In Table 3, we compare the single-domain baseline (on mini-ImageNet), the domain-mix and the proposed domain-switch scheme (without domain-general teacher and the domain-specific prompter ) on CUB and Cars. We draw two following observations:

First, compared with single-domain training, the domain-mix learning scheme only brings slight improvement. Although it employs multiple domains (as well as multiple datasets) for training, the benefit of more training data is trivial (less than 1% improvement). We infer it is because in the domain-mix learning scheme, the disadvantage of learning domain-specific knowledge largely offsets the advantage of more training data.

Second, comparing "domain-mix" with "domain-switch", we observe that domain-switch is a superior multi-domain learning manner. Specifically, domain-switch surpasses domain-mix by 1.60% accuracy under 5-way 5-shot setting on CUB. It is consistent with our conceptual analysis, *i.e.*, the domain-switch learning manner better ignores the domain-specific knowledge, therefore improving the cross-domain generalization.

## 4.4 ABLATION STUDIES

**Ablation on the prompter and teacher module.** While the domain-switch framework already improves cross-domain generalization, DSL uses the domain-specific prompter and the domain-general teacher to prevent over-fitting domain-specific knowledge and to reinforce the domain-

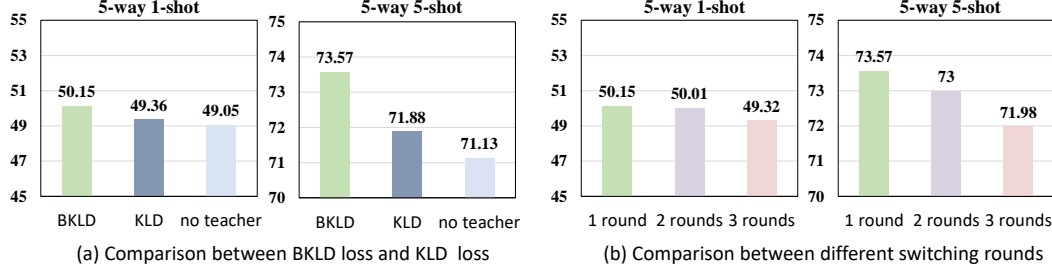

Figure 4: Analysis on Binary KLD loss and the number of switching round of teacher module.

| Components | 5-way 1-shot | 5-way 5-shot |
|---|---|---|
| domain-switch | $46.34 \pm 0.73\%$ | $66.33 \pm 0.71\%$ |
| domain-switch + specific prompter | $49.05 \pm 0.60\%$ | $71.13 \pm 0.68\%$ |
| domain-switch + general teacher | $48.88 \pm 0.61\%$ | $70.45 \pm 0.71\%$ |
| **All (DSL)** | **$50.15 \pm 0.80\%$** | **$73.57 \pm 0.73\%$** |

Table 4: Ablation studies of our proposed method on individual components.

general knowledge, respectively. We investigate their benefits through ablation in Table 4, from which we draw two observations:

*First, both the domain-specific prompter and the domain-general teacher are beneficial.* Based on the domain-switch operation, we find that adding the prompter or the teacher brings accuracy improvement. For example, under the 5-way 5-shot setting on CUB, adding the prompter and the teacher increases the accuracy by $4.80\%$ and $4.12\%$, respectively.

*Second, the domain-specific prompter and the domain-general teacher are complementary to each other.* While both modules already bring independent improvement, combining them ("All (DSL)") yields even higher accuracy. It validates that these two modules achieve complementary benefits (*i.e.*, countering domain-specific knowledge and promoting domain-general knowledge).

**Analysis on some important configurations of the teacher module.** We investigate two configurations: 1) using a novel Binary KLD loss (instead of the popular KLD loss) for knowledge distillation and 2) using the historical models within a single switching round for generating the teacher.

Fig. 4 (a) shows that the domain-general teacher favors the Binary KLD loss than the canonical KLD loss. Under the 5-way 5-shot setting on CUB, the canonical KLD barely improves over "no teacher", while the BKLD surpasses KLD by $1.69\%$ accuracy. We infer it is because the knowledge from other domains is not accurate for the non-target classes. Therefore, forcing the learner to approximate the teacher *w.r.t.* all the non-target classes is inappropriate.

Fig. 4 (b) shows that using more switching rounds gradually compromises the effectiveness of the teacher, compared with using a single round. We infer it is because increasing switching rounds includes some severely out-of-date models, whose knowledge largely diverges from the up-of-date model. Therefore, we use a single switching round for generating the teacher model.

## 5 CONCLUSION

This paper proposes a novel Domain-Switch Learning (DSL) method for cross-domain few-shot classification. DSL embeds the cross-domain scenario into the training stage in a fast switching manner. We show that during the domain switching procedure, the deep model favors domain-general knowledge and is prone to ignoring the domain-specific knowledge, so as to fast adapt itself to different domains. Moreover, DSL employs a domain-specific prompter and a domain-general teacher module to further promote the cross-domain generalization capacity. Experiments conducted on multiple benchmarks demonstrate that DSL improves cross-domain few-shot classification and the achieved results are on par with the state of the arts.

ETHICS STATEMENT

This paper can help to improve the model generalization capacity in recognizing novel classes with limited labeled samples (*e.g.,* the rare birds, the rare plants,etc) under a significant domain gap between training sets. It may be applied to some environmental protection projects which require monitoring on the diversity of species. Specifically, it can help recognize some rare species in the wild reserve. We will explore more application scenarios of few-shot learning, as well as cross-domain few-shot learning. Moreover, we will try to improve the reliability of few-shot learning systems to reduce the potential problems.

REPRODUCIBILITY STATEMENT

The DSL is reproducible. In the main text, we describe the utilized datasets in DSL, *i.e.,* 4 fine-grained datasets (CUB, Cars, Places, and Plantae) and 1 popular large scale dataset mini-ImageNet. We provide the details about the experimental implementation, the proof of the proposed remark and the analysis of some hyper parameters in appendix.

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

# A    APPENDIX

In the appendix, we supply the details which are not described in the main text due to space limitation. In Section A.1, we provide the details about the experimental implementation. In Section A.2, we prove the Remark2 in Section 3.3. In Section A.3, we investigate DSL on different backbones. In the main experiments, DSL and all the competing methods adopt the standard inductive inference. In Section A.4, we further provide some comparison under the transductive setup (*e.g.*, with a label propagation method  Liu et al. (2020)).  In Section A.5, we analyze the impact of some hyper-parameters.

## A.1    IMPLEMENTATION DETAILS.

We use ResNet-10 He et al. (2016) without pretraining as the backbone network. When training the model on the training set, we use the Adam Kingma & Ba (2015) optimizer. We train the model with 400 epochs and set the initial learning rate as 1e-3. In each training iteration, the mini-batch size of switchable domain and the "basic" training set are both 64. In each mini-batch , we randomly sample 16 classes from each domain (4 images per class).

We evaluate the classification accuracy over 1000 experiments on the test set. When fine-tuning the model on the support set, we randomly sample $C$ classes and $K$ samples per class, according to the $C$-way $K$-shot setting(*e.g., C*=5, *K*=1 or 5). We use the SGD Qian (1999) to optimize the linear classifier layer and the initial learning rate is 0.01. The weight decay of SGD is 0.001 and the SGD momentum is 0.9.

## A.2    REMARK2 PROOF.

We first recall the domain-specific prompter in Section 3.3. Given a sample $x$ in current domain, we average the softmax predictions of the prompter $P(y|x; \theta^{\mathcal{P}}, W^{\mathcal{P}})$ and learner $P(y|x; \theta, W)$ to get re-weighted softmax prediction $\hat{P}$. It can prevent the learner from over-fitting the current domain.

**Remark2**: Given $P(y|x; \theta^{\mathcal{P}}, W^{\mathcal{P}}) \geq P(y|x; \theta, W)$, we have the re-weighted softmax prediction $\hat{P} \geq P(y|x; \theta, W)$. It thus reduces final prediction errors (on the current domain), alleviating the penalty on the learner. The proof is as illustrated by:

$$
\begin{aligned}
&\frac{1}{P(y|x; \theta^{\mathcal{P}}, W^{\mathcal{P}})} \leq \frac{1}{P(y|x; \theta, W)} \\
\Rightarrow &\frac{1}{\lambda \cdot P(y|x; \theta^{\mathcal{P}}, W^{\mathcal{P}}) + (1-\lambda) \cdot P(y|x; \theta^{\mathcal{P}}, W^{\mathcal{P}})} \leq \frac{1}{P(y|x; \theta, W)} \\
\Rightarrow &\frac{\lambda}{\lambda \cdot P(y|x; \theta^{\mathcal{P}}, W^{\mathcal{P}}) + (1-\lambda) \cdot P(y|x; \theta^{\mathcal{P}}, W^{\mathcal{P}})} \leq \frac{1}{P(y|x; \theta, W)} \\
\Rightarrow &|\frac{\mathrm{d}\mathcal{L}_{RCE}}{\mathrm{d}\hat{P}}| \leq |\frac{\mathrm{d}\mathcal{L}_{CE}}{\mathrm{d}P(y|x; \theta^{\mathcal{P}}, W^{\mathcal{P}})}|
\end{aligned}
\tag{8}
$$

where $|\cdot|$ denotes the value of vector. $\lambda \in [0, 1]$ is a hyper parameter.

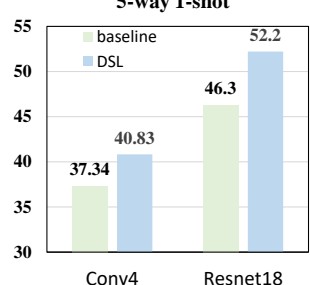
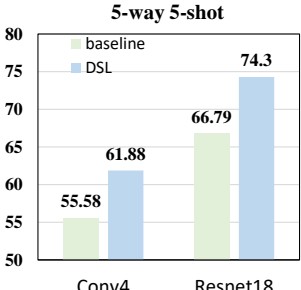

Figure 5:  Illustration of our proposed method on both shallow or deeper backbone settings.

## A.3 DSL ON VARIOUS BACKBONES.

We further analyze the adaptation of our methods on Conv4 and ResNet18. The performance on CUB are shown in Fig. 5. We adopt "domain-mix" learning manner as the baseline. We observe that the proposed method can still achieve considerable improvements on both shallow or deeper backbone settings, which proves the robustness of the proposed method.

## A.4 ADDITIONAL EXPERIMENTS.

We adopt a simple label propagation method Liu et al. (2020) for DSL and compare "DSL+LP" with label-propagation based methods.

| Methods | CUB | Cars | Places | Plantae |
|---------|-----|------|--------|---------|
| TPN+ATA | $65.31 \pm 0.40$ | $46.95 \pm 0.40$ | $72.12 \pm 0.40$ | $55.08 \pm 0.40$ |
| GNN | $69.26 \pm 0.68$ | $48.91 \pm 0.67$ | $72.59 \pm 0.67$ | $58.36 \pm 0.68$ |
| GNN+LFT | $73.11 \pm 0.68$ | $49.88 \pm 0.67$ | $77.05 \pm 0.65$ | $58.84 \pm 0.66$ |
| **DSL+ LP** | $\mathbf{76.72 \pm 0.60}$ | $\mathbf{62.21 \pm 0.70}$ | $\mathbf{77.10 \pm 0.70}$ | $\mathbf{65.70 \pm 0.75}$ |

Table 5: Comparison with the state of the arts for 5-way 5-shot task. "ATA" denotes Adversarial Task Augmentation,"LFT" denotes learning-to-learned feature-wise transformation.

From Table 5, we observe that when all the methods use transductive setup (*e.g.*, label propagation), DSL achieves competitive accuracy under the 5-way 5-shot classification. 'DSL+LP' outperforms GNN + LFT by +3.61%, +12.33%, +0.05%, +6.86% on CUB, Cars, Places and Plantae, respectively.

## A.5 HYPER-PARAMETERS ANALYSIS.

We analyze the impact of three important hyper-parameters,i.e.,$\lambda$ in re-predicted procedure(Eq. 3), the $\alpha$ in overall loss(Eq. 7) and the temperature coefficient $\mu$ in computing the predictions $P_y^{\mathcal{T}} = P(y|\boldsymbol{x}; \boldsymbol{\theta}^{\mathcal{T}}, \boldsymbol{W}^{\mathcal{T}})$ of domain-general teacher.

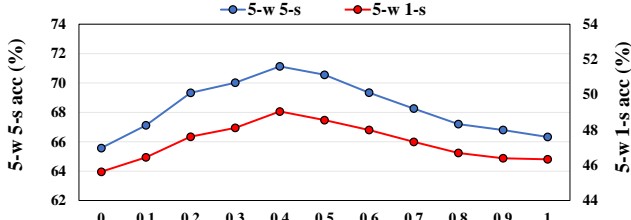

Figure 6: Analysis on the hyper-parameter $\lambda$.

In Fig. 6, we evaluate the impact of hyper-parameter $\lambda$, which controls the weight of predictions from domain-specific prompter in Eq. 3. We observe that the accuracy first increases (when $\lambda$ increases from 0 to 0.4) and then decreases (when $\lambda$ further increases to 1.0). We set $\lambda = 0.4$ as the weight factor.

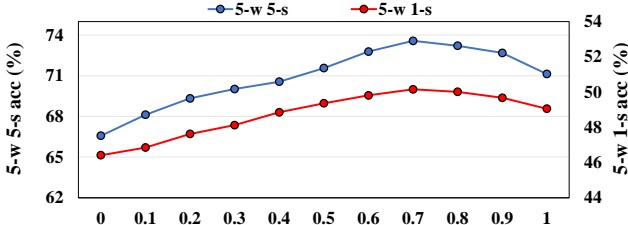

Figure 7: Analysis on the hyper-parameter $\alpha$.

In Fig. 7, we evaluate the impact of hyper-parameter $\alpha$, which controls the weight of Binary KLD loss and RCE loss in Eq. 7. We observe that the accuracy first increases (when $\alpha$ increases from 0 to 0.7) and then decreases (when $\alpha$ further increases to 1.0). We set $\alpha = 0.7$ as the weight factor.

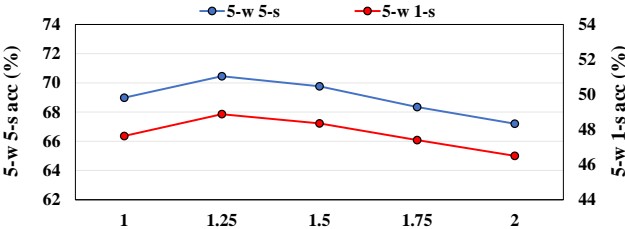

Figure 8: Analysis on the hyper-parameter $\mu$.

In Fig. 8, we evaluate the impact of hyper-parameter $\mu$, which denotes the temperature coefficient in computing predictions $P_y^{\mathcal{T}} = P(y|\boldsymbol{x}; \boldsymbol{\theta}^{\mathcal{T}}, \boldsymbol{W}^{\mathcal{T}})$ of domain-general teacher. When the temperature coefficient $\mu \geq 1$, we consider that the predictions of domain-general teacher are more smooth and thus transfer more general knowledge shared with other domains. We set $\mu$ to vary from 1 to 2. It is observed that the accuracy undergoes an increase (when $\mu$ increases from 1 to 1.25) and then a decrease (when $\mu > 1.25$). Therefore, we set $\mu = 1.25$ as the optimized threshold for domain-general teacher.

