# OpenReview forum: "Switch to Generalize: Domain-Switch Learning for Cross-Domain Few-Shot Classification"
_ICLR.cc/2022/Conference — ICLR 2022 Poster_

### Official Review · Reviewer_KRoA · 2021-11-02

**Correctness:** 3
**Technical Novelty And Significance:** 3
**Empirical Novelty And Significance:** 3
**Recommendation:** 8
**Confidence:** 3

**Main Review:**





**Strengths**
1. Overall, this paper is easy to follow. The introduction nicely outlines the method.
2. The domain-specific prompter and the domain-general teacher are straightforward and reasonable. The two modules are needed under the proposed domain-switch learning manner.
3. Domain-general teacher seems a general module that can be used in other few-shot frameworks (although the experiment does not validate this point).
4. The experimental evaluation nicely stuides the hyper-parameters.

**Weaknesses**

- **[Working mechanism needs more discussion]**

	The proposed Domain-Switch Learning (DSL) includes only a single domain into a training iteration and switches to another domain for the following iteration. The work highlights several times that such a way makes “the deep model favors domain-general knowledge and is prone to ignoring the domain-specific knowledge”. This point is not very clear and needs more validations.

	***First***, the domain-mix pipeline learns images from different domains in each iteration. In order to learn well from these images, the model tends to learn common information. This also makes the model favors domain-general knowledge. Thus, it is still not clear why domain-switch learning should be better than domain-mix learning.

	***Second***, In Sec. 3. 2 the work includes a sentence to eliminate the concern of the contradiction when using mini-ImageNet images at each iteration. However, it is still not very convincing.
	A. One could also regard each mixed batch in domain-mix learning as a single “compound” domain.
	B. Also, following the logic of the motivation, using mini-ImageNet in each iteration could make model memory the information of mini-ImageNet (which is domain-specific). Such mini-imagenet specific information might already be general, which is helpful when testing on bird/car/place images. This might be the reason why including mini-ImageNet images in each iteration helps to learn.

	***Third***, experimental comparison between domain-switch and domain-mix needs clarification (Please see the first point in the following).

- **[Experimental evaluation needs clarification]**

	***First***, Table 3 needs more clarification. 1) When reporting results on CUB, does the model use four domain images (mini-ImageNet, Car, Places, Plantae) for training? 2) when using the domain-mix manner, does each mini-batch contain mini-ImageNet images? 3) one possible baseline could be first pertaining on mini-ImageNet, and then finetuning using domain-mix manner or domain-switch manner. 4) do the methods listed in Tables 1 and 2 also use mini-ImageNet image at each training iteration or just finetuning?

	***Second***, Tables 1 and 2 list basic baselines. GNN (Tseng et al. (2020)) should be included as well. Moreover,  the results of NASE and BSR under the multi-domain setting (“m” in the table) are missing.

	***Third***, when using the teacher network, why BKLD is better (as shown in Fig4 (a))? It would be better if this work could provide a discussion on this.

- Minor:

	[-] The captions of Tables 1 and 2 are not clear. The definitions of TF, ATA, s, m are missing.
	[-] Summarization of the datasets (domains) used in the experiment could be included in the appendix.

***---Post Rebuttal---***

**First**, ***the major concern of "working mechanism" is well resolved by the clarification***. The authors tune down their claim and clarify that "The superiority of domain-switch (without teacher or prompter) is not because it learns the domain-general knowledge better. Instead, it is because domain-switch learner better suppresses the domain-specific knowledge". This is reasonable and much clearer.

***One additional comment is that the introduction does not fully highlight this point***. The corresponding text should be modified.

**Second**, ***the evaluation is well clarified.*** ***One suggestion*** is the main paper should clearly mention the experiments in the appendix. These results are helpful to understand the effectiveness of the proposed method.

In addition, I notice that the experimental results on the ***newly-introduced test sets (Aircraft and Traffic Sign) also show the effectiveness of DSL***. ***It would be better*** if the authors could add them in the revised paper (e.g., appendix), which helps understand the effectiveness of DSL.

Based on the above points, I am inclined to accept this paper.

**Summary Of The Paper:**

The goal of this paper is to improve the generalization ability of the few-shot model on novel domains. To this end, this work introduces a learning manner named domain-switch learning (DSL). To make DSL feasible, the work further proposes two modules (i.e., domain-specific prompter and the domain-general teacher). The experimental results show the effectiveness of the proposed framework.

**Summary Of The Review:**

This work bases on the domain switch learning (DSL) manner. Around this, the work proposes two modules (i.e., domain-specific prompter and the domain-general teacher). Two modules are reasonable and needed under the DSL.

The major concern is the working mechanism of DSL is not well explained. More discussion/explanation could help eliminate the concern.  Moreover, experimental evaluation needs clarification.

***---Post Rebuttal---*** After reading the response and revised paper, my major concerns are well addressed by the authors.

---

> ### Author Response · Authors · 2021-11-15
> **Reply to Reviewer KRoA (1/2)**
>
> Thanks for your comments and positive feedback! We clarify the mentioned concerns in more details and hope the reviewer will raise final rating.
>
> Q1: First, the domain-mix pipeline learns images from different domains in each iteration. In order to learn well from these images, the model tends to learn common information. This also makes the model favors domain-general knowledge. Thus, it is still not clear why domain-switch learning should be better than domain-mix learning.
>
> Ans: Thanks for the question. The superiority of domain-switch (without teacher or prompter) is not because it learns the domain-general knowledge better. Instead, it is because domain-switch learner better suppresses the domain-specific knowledge.
>
> As explained to Reviewer JgKH-Q2, when we stated that "the domain-switch favors the domain-general knowledge and is prone to ignore the domain-specific knowledge", it might make one feel that domain-switch learns domain-general knowledge better than the domain-mix does. We apologize for this unclear statement and will revise it for better understanding. The reality is that domain-switch suppresses the domain-specific knowledge at the cost of losing some domain-general knowledge. Domain-general teacher makes some compensation for this loss and thus brings improvement to DSL.
>
> Q2: Second, In Section 3.2 the work includes a sentence to eliminate the concern of the contradiction when using mini-ImageNet images at each iteration. However, it is still not very convincing. A. One could also regard each mixed batch in domain-mix learning as a single “compound” domain. B. Also, following the logic of the motivation, using mini-ImageNet in each iteration could make model memory the information of mini-ImageNet (which is domain-specific). Such mini-imagenet specific information might already be general, which is helpful when testing on bird/car/place images. This might be the reason why including mini-ImageNet images in each iteration helps to learn.
>
> Ans: We would like to provide further explanations on this intuition and hope it will address your concern.
> As described in the 4-th paragraph in introduction, we consider each domain contains both domain-general and domain-specific knowledge. Arguably, if a dataset contains infinite number of samples, it is likely that this dataset has no more domain bias and contains no more domain-specific knowledge. Analogically, we note that mini-ImageNet is a large-scale dataset and is largely dominated by domain-general knowledge.
>
> Therefore, if we use mini-ImageNet as a switchable dataset, the disadvantage of forgetting some domain-general knowledge outperforms the benefits of suppressing domain-specific knowledge. Moreover, using mini-ImageNet as a basic training set is consistent with the common practice in cross-domain few-shot learning community.
>
> Q3: Table 3 needs more clarification. 1) When reporting results on CUB, does the model use four domain images (mini-ImageNet, Car, Places, Plantae) for training? 2) when using the domain-mix manner, does each mini-batch contain mini-ImageNet images? 3) one possible baseline could be first pertaining on mini-ImageNet, and then finetuning using domain-mix manner or domain-switch manner. 4) do the methods listed in Tables 1 and 2 also use mini-ImageNet image at each training iteration or just finetuning?
>
> Ans: Thanks for your suggestions, we would like to answer these question point by point and will add the explanations to the manuscript for clarity.
> 1) Yes, Table 3 uses four domain images for training.
> 2) Yes, the domain-mix manner includes mini-ImageNet into every mini-batch for training.
> 3) Good suggestion. Pre-training with mini-ImageNet and then fine-tuning with three switchable domains achieves lower accuracy on CUB (5-way 5-shot) These two results are significantly lower than domain-switch and domain-mix by -6.12% and -6.78%, respectively.
> 4) Yes, all the compared methods in Table 1 and 2 use mini-ImageNet as their only or major training set.

---

> ### Author Response · Authors · 2021-11-15
> **Reply to Reviewer KRoA (2/2)**
>
> Q4: Tables 1 and 2 list basic baselines. GNN (Tseng et al. (2020)) should be included as well. Moreover, the results of NASE and BSR under the multi-domain setting (“m” in the table) are missing.
>
> Ans: thanks for your suggestion. We did not compare DSL against GNN in the manuscript, because GNN adopts label propagation in testing stage, while all the compared methods (including our DSL) are inductive and do not use label propagation. That being said, we provide the comparison with GNN method for rebuttal. The results are shows DSL
>
> | Methods | CUB | Cars | Places | Plantae |
> | --- | --- | --- | --- | --- |
> |GNN + LFT (m)| 73.11 | 49.88 | 77.05 | 58.84 |
> |DSL + LP (m)| 76.72 | 62.21 | 77.10 | 65.70 |
> where 'LP' represents label propagation proposed in [e].
>
> Under the 5-way 5-shot setting, 'DSL+LP [e]' outperforms GNN+LFT by +3.61%, +12.33%, +0.05%, +6.86% on CUB, Cars, Places and Plantae, respectively. Meanwhile, the authors of NASE do not report the performance under multi-domain setting and have not released their code either. Therefore, we use our implementation for comparison. Under 5-way 5-shot setting, NASE + BSR(m) achieves 72.10 on CUB and is lower than DSL+BSR(m) by -2.18%.
>
> Q5: When using the teacher network, why BKLD is better (as shown in Fig4 (a))? It would be better if this work could provide a discussion on this.
>
> Ans: We apologize for having not highlighted the discussion. We actually already provided such a discussion in Section 4.4 (Fig. 4 (a)):
> "Under the 5-way 5-shot setting, the canonical KLD barely improves over “no teacher”, while the BKLD surpasses KLD by 1.69% accuracy. We infer it is because the knowledge from other domains is not accurate for the non-target classes. Therefore, forcing the learner to approximate the teacher w.r.t. all the non-target classes is inappropriate."
>
> Q6: The captions of Tables 1 and 2 are not clear. The definitions of TF, ATA, s, m are missing. [-] Summarization of the datasets (domains) used in the experiment could be included in the appendix.
>
> Ans: Thanks for your suggestion. We will clarify the captions for Tables 1 and 2.
> 1) 'FT' means pre-determined feature-wise transformation in [i]
> 2) 'LFT' means learning-to-learned feature-wise transformation in [i]
> 3) 'ATA' means Adversarial Task Augmentation in [a]
> 3) 's' means single-domain learning scheme
> 4) 'm' means multi-domain learning scheme
> 5) We will discuss the datasets in the appendix.
>
> [a] Haoqing Wang and Zhi-Hong Deng. Cross-domain few-shot classification via adversarial task augmentation. arXiv preprint arXiv:2104.14385, 2021.
>
> [e] Bingyu Liu, Zhen Zhao, Zhenpeng Li, Jianan Jiang, Yuhong Guo, and Jieping Ye. Feature transformation ensemble model with batch spectral regularization for cross-domain few-shot classification. arXiv preprint arXiv:2005.08463, 2020.
>
> [i] Hung-Yu Tseng, Hsin-Ying Lee, Jia-Bin Huang, and Ming-Hsuan Yang. Cross-domain few-shot classification via learned feature-wise transformation.ICLR 2020.

---

> ### Author Response · Authors · 2021-11-30
> **Reply to Reviewer KRoA (post rebuttal)**
>
> We appreciate that Reviewer KRoA considers our responses informative and raises his / her rating to "Accept". We will further polish our introduction w.r.t. the comparison between domain-switch and domain-mix scheme. We will also clearly mention the appendix experiments in the main paper. Thanks for all these suggestions.

---

### Official Review · Reviewer_gTju · 2021-11-02

**Correctness:** 3
**Technical Novelty And Significance:** 2
**Empirical Novelty And Significance:** 2
**Recommendation:** 5
**Confidence:** 4

**Main Review:**

Paper Strengths:

The authors tackle an important and challenging problem of cross-domain few-shot classification. The proposed approach is simple. Experimental evaluations clearly demonstrate the effect by introducing the domain switching training strategy.

Paper Weaknesses:

1) The approach is evaluated on CUB, Cars, Places, and Plantae in a leave-one-out manner. These four datasets are all fine-grained and consist of natural images. I was wondering how does the proposed domain-switch strategy work on more diverse, heterogenous datasets, like Meta-dataset [Meta-Dataset: A Dataset of Datasets for Learning to Learn from Few Examples, ICLR 2020].

2) The comparisons in the paper are a little bit weak. It is only marginal better than NASE. How is the performance of NASE in the multi-domain setting? Also, comparisons with some other existing cross-domain few-shot learning methods are missing, for example [Meta-Dataset: A Dataset of Datasets for Learning to Learn from Few Examples, ICLR 2020] [Selecting Relevant Features from a Multi-domain Representation for Few-shot Classification, ECCV 2020] [A Universal Representation Transformer Layer for Few-Shot Image Classification, ICLR 2021].

3) In Remarks 1, 2, and 3, the relationships between the predicted probability of the prompter, the learner, and the teacher are discussed. While the reasoning intuitively makes sense, there is no mathematically rigorous proof. The current statements sound like they are already-proved facts.

4) While in the introduction it is stated that the model is only trained on one domain at each iteration, one critical implementation detail is that the approach requires a large-scale basic training set (mini-ImageNet) to stabilize the training procedure. I think this should be mentioned in the introduction. Moreover, an ablation study should be provided – what is the performance without mini-ImageNet? Also, if the model is pre-trained on mini-ImageNet, and then trained on the individual domain in a domain-switch manner, does that work?

5) Currently, the training is conducted periodically with pre-defined domain order. How is the sensitivity to the domain order? How is the performance if the domain ordering is completely random in the entire training procedure? In that case, probably the design of the prompter and teacher should be modified as well.

6) Domain-mix and domain-switch can be viewed as two extremes, where domain-mix uses all the domains at each iteration, while domain-switch uses only one domain at each iteration. How is the performance if we use some strategy in between? For example, at each iteration, some domains are randomly combined as a compound domain, and there is no overlapping domain between two consecutive training iterations.

**Summary Of The Paper:**

This paper aims to address the problem of cross-domain few-shot classification. The main contribution is that it introduces domain-switch learning that simulates cross-domain scenario for training, and thus improves the model’s generalization. To this end, it uses multiple domains during training – it only trains on a single domain at each iteration and switches the domain in consecutive training iterations. This training mechanism is different from widely-used strategies like mixing different domains, and it forces the model to learn domain-general knowledge. In addition, two constraints are proposed to further improve the generalization. First, a domain-specific prompter module is introduced, so that the model is not overfitting to the domain in the current iteration. Second, a domain-general teacher module is introduced, so that the model does not forget the already-learned knowledge of other domains. The approach is tested on four datasets, including CUB, Cars, Places, and Plantae, and compared with competing results.

**Summary Of The Review:**

The paper introduces domain-switch learning that simulates cross-domain scenario for training, and thus improves the model’s generalization. The evaluations and comparisons in the paper are a little bit weak. Some ablation studies and analysis are missing.

---

> ### Author Response · Authors · 2021-11-15
> **Reply to Reviewer gTju (1/2)**
>
> Thanks for your careful comments. We clarify the mentioned concerns in more details and hope the reviewer will raise final rating.
>
> Q1: The approach is evaluated on CUB, Cars, Places, and Plantae in a leave-one-out manner. These four datasets are all fine-grained and consist of natural images. I was wondering how does the proposed domain-switch strategy work on more diverse, heterogenous datasets, like Meta-dataset [Meta-Dataset: A Dataset of Datasets for Learning to Learn from Few Examples, ICLR 2020].
>
> Ans: Our choice of datasets is standard in cross-domain few-shot learning and is consistent with all the competing methods in Table 1 and Table 2 in the manuscript. We feel a bit confused on the suggestion of  using Meta-Dataset for evaluation, because the underlying tasks of two kinds of datasets are fundamentally different. Specifically, Meta-dataset is for multi-domain few shot learning, where the testing domain has no domain labels but is already seen during training. This is different from the cross-domain few shot learning, where the testing domain is unseen and novel. It is possible that we have missed something from your suggestion, so please let us know if you have detailed questions.
>
> Q2: The comparisons in the paper are a little bit weak. It is only marginal better than NASE. How is the performance of NASE in the multi-domain setting? Also, comparisons with some other existing cross-domain few-shot learning methods are missing, for example [Meta-Dataset: A Dataset of Datasets for Learning to Learn from Few Examples, ICLR 2020] [Selecting Relevant Features from a Multi-domain Representation for Few-shot Classification, ECCV 2020] [A Universal Representation Transformer Layer for Few-Shot Image Classification, ICLR 2021].
>
> Ans: First, we beg to differ from the statement that "the comparisons are a little bit weak". The compared methods in the manuscript already cover majority of the cross-domain few-shot learning literature.
>
> Second, the authors of NASE do not report the performance under multi-domain setting and have not released their code either. Therefore, we use our implementation for comparison. We find that under the 5-way 5-shot setting, NASE(m) achieves 69.82 on CUB, which is lower than DSL by -3.75%.
>
> Third, the last 2 methods recommended by reviewer are not quite comparable to DSL, because they are all multi-domain few-shot learning methods. As stated in Q1, multi-domain FSL is fundamentally different to cross-domain FSL.
>
> Q3: In Remarks 1, 2, and 3, the relationships between the predicted probability of the prompter, the learner, and the teacher are discussed. While the reasoning intuitively makes sense, there is no mathematically rigorous proof. The current statements sound like they are already-proved facts.
>
> Ans: Thank you for recognizing our reasoning as being intuitive and rational. We have to admit that many phenomena in deep learning still have no rigorous mathematical proof. That being said, we would like to provide some further discussions on these Remarks.
>
> 1) Remark 1 is consistent with the well-known 'catastrophic forgetting' [f][g][h] in deep learning. According to the catastrophic forgetting phenomenon, when the deep model is trained sequentially on multiple tasks (domains), there is a tendency that the knowledge learned on former tasks (domains) is abruptly lost. In our case, the domain-specific prompter is the former model updated by Di and thus fits Di well. In contrast, the current learner is newly updated by a different Dj and thus already forgets some knowledge of Di. Therefore, the prompter has higher predicted probability on the ground-truth category than the learner does. Meanwhile, When the learner gets updated from Di, it has higher prediction accuracy on Di than the teacher does but has lower prediction accuracy on former domains due to
>
> 2) We already provided the proof of Remark 2 in Appendix (A.2) in the manuscript due to the space limited.
>
> 3) Remark 3 is empirically validated by Fig.3(b) in the manuscript.
>
> [f] James Kirkpatrick, Razvan Pascanu, Neil Rabinowitz, Joel Veness, Guillaume Desjardins, Andrei A Rusu, Kieran Milan, John Quan, Tiago Ramalho, Agnieszka Grabska-Barwinska, et al. Overcoming catastrophic forgetting in neural networks. Proceedings of the national academy of sciences, 114(13):3521–3526, 2017.
>
> [g] Michael McCloskey and Neal J Cohen. Catastrophic interference in connectionist networks: The sequential learning problem. In Psychology of learning and motivation, volume 24, pp. 109–165. Elsevier, 1989.
>
> [h] Robert M French. Catastrophic forgetting in connectionist networks. Trends in cognitive sciences, 3(4):128–135, 1999.

---

> ### Author Response · Authors · 2021-11-15
> **Reply to Reviewer gTju (2/2)**
>
> Q4: While in the introduction it is stated that the model is only trained on one domain at each iteration, one critical implementation detail is that the approach requires a large-scale basic training set (mini-ImageNet) to stabilize the training procedure. I think this should be mentioned in the introduction. Moreover, an ablation study should be provided – what is the performance without mini-ImageNet? Also, if the model is pre-trained on mini-ImageNet, and then trained on the individual domain in a domain-switch manner, does that work?
>
> Ans: Thanks, we will mention in the introduction that mini-ImageNet is used as a basic training set. We also point out that all the previous methods use the mini-ImageNet dataset as their basic training set:
> 1) Single-domain training methods only adopt mini-ImageNet as the source domain.
> 2) Previous multi-domains training method [i] adopts mini-ImageNet as the basic domain during every training stage, as well.
>
> Therefore, using the mini-ImageNet as the basic training set is consistent with the common practice in cross-domain few-shot learning community, while removing mini-ImageNet during training is rare. That being said, during rebuttal, we provide two experimental comparisons according to your questions as below:
>
> 1) If we remove mini-ImageNet, we achieve only 62.46 accuracy on CUB under 5-way 5-shot setting, which is lower than DSL by -11.11%.
>
> 2) If we use mini-ImageNet only for pre-training, we achieve only 64.13 accuracy on CUB under 5-way 5-shot setting, which is lower than DSL by -9.44%.
>
> Q5: Currently, the training is conducted periodically with pre-defined domain order. How is the sensitivity to the domain order? How is the performance if the domain ordering is completely random in the entire training procedure? In that case, probably the design of the prompter and teacher should be modified as well.
>
> Ans: Thanks for the question. DSL has no special preference on any specific order, but requires that in any single switching round, each switchable domain appears exactly once (according to the definition of switching round in Section 3.2 in the manuscript). This prerequisite is to ensure that the domain-general teacher can absorb knowledge from all the domains. Any specific order arrangement (e.g., "[1-2-3]-[1-2-3]...", "[1-3-2]-[1-3-2]...") satisfying this prerequisite will be good for DSL.
>
> If the domain ordering is completely random, the achieved accuracy decreases. For example, on 5-way 5-shot CUB, it is lower than DSL by by about -2.1\%. The reason is that the random order contradicts the definition of switching round. Consequently, some domain X may be absent from training for a relatively long period, making us use severely out-of-date model as the prompter for domain X. Meanwhile, some other domain Y may appear more than once in a switching round, making the teacher biased towards Y.
>
> Q6: Domain-mix and domain-switch can be viewed as two extremes, where domain-mix uses all the domains at each iteration, while domain-switch uses only one domain at each iteration. How is the performance if we use some strategy in between? For example, at each iteration, some domains are randomly combined as a compound domain, and there is no overlapping domain between two consecutive training iterations.
>
> Ans: Thanks for your suggestion. Randomly combining domains into a compound domain also incurs the problem of contradicting the definition of switching round. Therefore, we find that it compromises the accuracy. There are two reasons:
> 1) It influences the effectiveness of domain-general teacher which aims to absorbs the average knowledge of all domains.
> 2) It harms the specific knowledge of domain-specific prompter if the random domains are combined in each training iteration.
>
> For example, on 5-way 5-shot CUB, it is lower than DSL by about -2.5%.
>
> [i] Hung-Yu Tseng, Hsin-Ying Lee, Jia-Bin Huang, and Ming-Hsuan Yang. Cross-domain few-shot classification via learned feature-wise transformation.ICLR 2020.

---

> ### Author Response · Authors · 2021-11-18
> **A New Response to Reviewer gTju Q1.**
>
> After re-checking your suggestion in Q1, we come into a new understanding. We guess you were asking us to use Meta-Dataset to construct the cross-domain few-shot learning scenario (instead of the original multi-domain few-shot learning task). If our guess is correct, we are able to evaluate the proposed DSL under heterogenous datasets. Therefore, without loss of generality, we choose three datasets (Birds, Omniglot and Textures) from Meta-Dataset (along with the mini-ImageNet) for training, and choose Aircraft and Traffic Signs for testing.
>
> We compare the "domain-mix", "domain-switch" and the DSL under the 5-way 5-shot setting as below. We observe that the domain-switch scheme surpasses the domain-mix counterpart (e.g., + 2.15% accuracy on Aircraft), and DSL further increases the accuracy to 60.16% (+9.19%). This observation validates the effectiveness of DSL on heterogenous datasets.
>
> | Methods | Aircraft | Traffic Signs |
> | --- | --- | --- |
> |domain-mix (multi-domain)| 50.97 | 73.17 |
> |domain-switch (multi-domain)| 53.12 | 76.08 |
> |DSL (multi-domain)| 60.16 | 80.33 |

---

> ### Comment · Reviewer_gTju · 2021-11-30
> **post rebuttal**
>
> I do appreciate the efforts that the authors made in the rebuttal. After reading the other reviewers’ comments and the authors’ rebuttal, my main concerns still remain, so I keep my original rating.
>
> The main question -- is the domain-switch strategy really better than the domain-mix strategy? From the paper and the rebuttal, it seems that the domain-switch strategy is kind of restrictive and not robust – “it requires single switching round, each switchable domain appears exactly once”; and requires mini-ImageNet to stabilize the training. And it seems that any slight modification to this strategy dramatically decreases the performance.
>
> In addition, the evaluation is constrained to a collection of somewhat homogenous datasets, like fine-grained datasets. That’s why initially I suggest evaluating on a collection of more heterogenous datasets, like Meta-dataset. Also, I kindly disagree with the authors that Meta-dataset is fundamentally different from the benchmark used in the paper. There are also fully held-out datasets like Traffic Signs and MSCOCO for evaluation on Meta-dataset. Basically, it is more convincing to show domain-switch is better than domain-mix at scale and with a greater number of domains and under more heterogenous setting. (I thank the authors for providing some partial results on Meta-dataset.)
>
> Second, I am a little bit concerned that the proposed approach is not compared against the strongest possible domain-mix method. And some of these methods are proposed on Meta-dataset. Making such comparisons will make the paper more convincing.
>
> In addition, the authors explain the source of performance improvement of the proposed approach comes from better suppression of the domain-specific knowledge. It would be more convincing to include more in-depth analysis to validate this.

---

> > ### Author Response · Authors · 2021-11-30
> > **Response to Reviewer gTju (post rebuttal)**
> >
> > We thank Reviewer gTju for the post-rebuttal comments. However, we respectfully disagree with some statements and these are explained below:
> >
> > **Comment: "the domain-switch strategy is kind of restrictive and not robust......any slight modification to this strategy dramatically decreases the performance"**.
> >
> > **Ans:** the switching strategy is not restrictive. "[1-2-3]-[1-2-3]", "[1-3-2]-[1-3-2]" and many other round-robin orders are good for DSL, as long as they enumerate all the switchable domains once in a single switching round.  This round-robin manner is simple and intuitive. It ensures that any two neighboring iterations have different domains and each domain is trained equally.
> >
> > In contrast, the random order strategy suggested in your initial comment **breaks the switching effect** because two neighboring iterations can share a same domain. Therefore, we kindly remind that such random order strategy **is not any slight modification of the switching strategy**.  The inferiority of random order strategy is actually an evidence showing the effectiveness of the proposed domain-switching strategy.
> >
> > As for using the mini-ImageNet, we clearly explained during the earlier response that it is a standard setup in all the previous cross-domain few-shot learning methods: 1) all previous multi-domain methods use mini-ImageNet as the base training set and 2) all single-domain methods use mini-ImageNet as the training set. Therefore, using mini-ImageNet is not a weakness but a standard choice.
> >
> > **Comment: "the evaluation is constrained to a collection of somewhat homogenous datasets......"**
> >
> > **Ans:** We already provide experiments on 3 (training) + 2 (testing) heterogenous datasets, apart from the fine-grained datasets (which is actually a common setup in previous cross-domain few-shot learning literature). We think these experiments are sufficient for validating the effectiveness of our method. We hope the reviewers and AC panel will appreciate our additional efforts under this new setup (which has seldom been explored in previous cross-domain few-shot learning literature).
> >
> > **Comment: "the proposed approach is not compared against the strongest possible domain-mix method. And some of these methods are proposed on Meta-dataset."**
> >
> > **Ans:** We already provide comparison against the strongest competitors including NAE and GNN (which is actually transductive). Generally, cross-domain few-shot learning methods and those multi-domain methods proposed on Meta-Dataset are not quite comparable. It is because the latter methods aim to robustify the deep model to **already-seen** domains (without domain label), while cross-domain few-shot learning methods aim to robustify the deep model to **unseen** domains.  No previous literature has ever conducted such comparison, to the best of our knowledge.
> >
> > That being said, some experiments in our rebuttal show that the proposed DSL is competitive, compared with multi-domain learning methods proposed on Meta-Dataset. For example, on Traffic Signs (a held-out dataset on Meta-Dataset), the proposed DSL achieves 80.33% accuracy (using only 3 training domains), outperforming two state-of-the-art methods SUR (70.4%) and URT (69.4%) by a clear margin. However, we would like to emphasize again that these methods (SUR and URT) mainly aim to robustify the deep model to **already-seen** domains and thus have a different task-of-interest from ours. The superiority of DSL on unseen domain does not mean that DSL is better than these methods under their own task-of-interest.

---

### Official Review · Reviewer_JgKH · 2021-11-03

**Correctness:** 3
**Technical Novelty And Significance:** 2
**Empirical Novelty And Significance:** 2
**Recommendation:** 6
**Confidence:** 3

**Main Review:**

Strength:
* the high-level intuition of incorporating adaptability into the training scheme is heuristic to the research community
* the proposed method is applicable to the real-life scenario of few-shot domain shift
* extensive experiments and ablation studies are conducted to demonstrate the performance of the proposed method
* the paper is very well written, easy to understand, well organized, and supported with nicely-drawn figures

Weakness:
* since the proposed method strictly requires multiple source domain, it is confusing how single domain baselines are relevant here. (it is included in all the experiment tables)
* the authors should consider adding more discussion on how the proposed method of domain-switch is better at extracting general information than domain-mix methods. Since in the domain-mix setting, different domains are given, it can be easier for the model to learn the general information shared by these domains, which is not possible in the domain switch setting.
* comparison with GNN would be helpful



**Summary Of The Paper:**

The authors propose to improve domain generalization capability by training with a fast switching manner. The intuition is that when the model is trained with the capability to adapt and retain information, such ability can be extended to adapt to a different test time distribution.

**Summary Of The Review:**

based on the above discussion, I recommend borderline accept for this paper. I would like to adjust my rating for the paper after discussing with the other reviewers and AC.

---

> ### Author Response · Authors · 2021-11-15
> **Reply to Reviewer JgKH**
>
> Thanks for your comments and positive feedback! We clarify the mentioned concerns in more details and hope the reviewer will raise final rating.
>
> Q1: Since the proposed method strictly requires multiple source domain, it is confusing how single domain baselines are relevant here. (it is included in all the experiment tables)
>
> Ans: It is because the superiority of multi-domain training against single-domain training is relatively trivial. For example, under the 5-way 5-shot setting, multi-domain only surpasses the single-domain baseline by +0.97%, +0.57% on CUB and Cars, respectively. We consider that such trivial superiority should not prohibit comparison against multi-domain and single-domain methods. Please refer to Reviewer (xu2A)-Q2 for more detailed discussions.
>
> Q2: The authors should consider adding more discussion on how the proposed method of domain-switch is better at extracting general information than domain-mix methods. Since in the domain-mix setting, different domains are given, it can be easier for the model to learn the general information shared by these domains, which is not possible in the domain switch setting.
>
> Ans: Thanks for this good suggestion. The comparison between domain-switch and domain-mix deserves further discussions. However, we would like to clarify a slight misunderstanding first. Although domain-switch scheme is better than domain-mix, it does NOT mean domain-switch has better capacity of extracting domain-general knowledge than the domain-mix scheme does.
> Instead, the superiority of domain-switch (without teacher or prompter) is mainly because domain-switch has better capacity to suppress the domain-specific knowledge (as introduced in Fig.1 in the manuscript). We explain the details as below:
>
> 1) We find our statement in the manuscript i.e., "the domain-switch favors the domain-general knowledge and is prone to ignoring the domain-specific knowledge" might be the reason for your misunderstanding and we will revise it for clarification later. This statement does not imply that domain-switch learns better domain-general knowledge than the domain-mix learner does. It is quite possible that domain-switch suppresses the domain-specific knowledge at the cost of losing some domain-general knowledge. Therefore, we agree with you on that "it is easier for domain-mix learns the general information shared by these domains".
>
> 2) However, in domain-switch, the benefit of suppressing the domain-specific knowledge dominates, and thus makes domain-switch better. Importantly, in DSL, the loss of domain-general knowledge can be compensated by the domain-general teacher. This is the reason why domain-general teacher substantially improves domain-switch, but barely improves domain-mix (see R(xu2A)-Q3 above).
>
> In overall, we find this suggestion is helpful for us to better compare domain-switch against domain-mix. It also helps us find a better viewpoint to present the motivation of the domain-general teacher (i.e., to compensate for the loss of domain-general teacher).
>
> Q3: Comparison with GNN would be helpful.
>
> Ans: Thanks for your suggestion. We did not compare DSL against GNN in the manuscript, because GNN adopts label propagation in testing stage, while all the compared methods (including our DSL) are inductive and do not use label propagation. During rebuttal, we add a comparison and find that DSL (assisted with a simple label propagation [e]) achieves higher accuracy than GNN. Under the 5-way 5-shot setting, 'DSL+LP [e]' outperforms GNN + LFT by +3.61%, +12.33%, +0.05%, +6.86%on CUB, Cars, Places and Plantae, respectively.
>
> | Methods | CUB | Cars | Places | Plantae |
> | --- | --- | --- | --- | --- |
> |GNN + LFT (m)| 73.11 | 49.88 | 77.05 | 58.84 |
> |DSL + LP (m)| 76.72 | 62.21 | 77.10 | 65.70 |
> where 'LP' represents label propagation.
>
> [e] Bingyu Liu, Zhen Zhao, Zhenpeng Li, Jianan Jiang, Yuhong Guo, and Jieping Ye. Feature transformation ensemble model with batch spectral regularization for cross-domain few-shot classification. arXiv preprint arXiv:2005.08463, 2020.

---

### Official Review · Reviewer_xu2A · 2021-11-03

**Correctness:** 3
**Technical Novelty And Significance:** 4
**Empirical Novelty And Significance:** 3
**Recommendation:** 6
**Confidence:** 4

**Main Review:**

Strengths
+ The paper is well written and easy to follow.
+ The proposed switch learning schedule seems novel for the cross-domain few-shot task.
+ The proposed Binary KLD Loss and Prompter loss seem to contribute a lot to the accuracy improvement.
+ The overall framework is consistent and the accuracies on standard benchmarks are acceptable.

Weaknesses
- According to Table1 and Table2, the authors directly copied the performance of RelationNet_ATA from [a], but why did not compare the performance of TPN+ATA. Compared with TPN+ATA, the strength of the proposed method would be limited, especially for the Places dataset. Could the authors give some explanation and analysis about that?

- Actually, the Tabel3 made me confused about the experimental results. (1) As mentioned in the first paragraph of Sec.4.2, the authors adopted the leave-one-out-setting, does it means the total number of training samples for multi-domain is larger than that for the single-domain? How to ensure the fairness of comparison？ （2）In Table3, I can't align the performance of the domain-mix scheme with other methods. The performance shown in Table3 has already been higher than some baseline models in Table1 and Table2, why?

- As shown in Table3, even though the accuracy of domain-mix was lower than domain-switch, I am still curious about the performance of domain-mix+teacher-regularization. In such a case, you could also use the average of several adjacent models as the teacher network to do the knowledge distillation. Further, if we use the teacher network to calculate both BKLD loss and RCE loss for the domain-mix scheme, how about the performance?

[a] Cross-Domain Few-Shot Classification via Adversarial Task Augmentation

**Summary Of The Paper:**

This paper proposes a novel training scheme termed Domain Switch Learning (DSL) to pursue the few-shot learning problem under the cross-domain scenario. In practice, DSL uses the data from one domain for a training iteration and switches to another domain for the next iteration. During the switching, two regularization terms are also introduced to balance the domain-specific knowledge and domain-general knowledge. The extensive experimental result demonstrates the effectiveness of the proposed method on several standard benchmarks.

**Summary Of The Review:**

For weaknesses 1 and 2, I hope the author could provide more detailed explanations.
For the last one, I am still concerned if the performance gain is provided by the switch scheme or just from the carefully designed BKLD loss and RCE loss.

---

> ### Author Response · Authors · 2021-11-15
> **Reply to Reviewer xu2A**
>
> Thanks for your comments and positive feedback! We clarify the mentioned concerns in more details and hope the reviewer will raise final rating.
>
> Q1: Why did not compare the performance of TPN+ATA? Could the authors give some explanation and analysis about that?
>
> Ans: Thanks for your question. We did not compare DSL against TPN+ATA, because TPN [b] is a transductive method, while the proposed DSL uses inductive inference. Since a transductive inference make additional use of the query and usually achieves higher accuracy than an inductive one (based on the same baseline), directly comparing their accuracy is not quite fair. Therefore, in the manuscript, we only compared the proposed DSL against ATA [a] based on RelationNet (which is also a inductive method). That being said, we find that DSL (inductive) actually presents considerable superiority against TPN (transductive) + ATA, i.e., DSL outperforms TPN+ATA by +8.26%, +11.58%, +1.98%, 7.02% on CUB, Cars, Places and Plantae, respectively.
>
> Q2: Actually, the Tabel3 made me confused about the experimental results.
>
> (1) As mentioned in the first paragraph of Sec.4.2, the authors adopted the leave-one-out-setting, does it means the total number of training samples for multi-domain is larger than that for the single-domain?How to ensure the fairness of comparison?
>
> Ans: Yes, the multi-domain setup employs more training samples than the single-domain setup. However, it does not impact the fairness in our experiments, because we find that multi-domain only brings trivial improvement over single-domain (under the domain-mix scheme). For example, under the 5-way 5-shot setting, multi-domain only surpasses the single-domain baseline by +0.97%, +0.57% on CUB and Cars, respectively. Given two facts that: 1) DSL improves the domain-mix baseline by a clear margin (8.84% and 7.33% on CUB and Cars, respectively), and 2) DSL surpasses the most competitive single-domain method (NASE) by a clear margin (5.06% and 3.62% on CUB and Cars, respectively), we find that the improvement due to using more training samples (less than 1%) is relatively trivial.
>
> (2) In Table3, I can't align the performance of the domain-mix scheme with other methods. The performance shown in Table3 has already been higher than some baseline models in Table1 and Table2, why?
>
> Ans: The domain-mix scheme indeed achieves higher accuracy than some baseline models in Table 1 and Table 2. It is because in this paper, we choose to use the fine-tuning pipeline, which generally gains higher performance than some meta-learning methods [c][d]. While the domain-mix scheme (based on the fine-tuning pipeline) is already very competitive, the improvement brought by the proposed DSL is still impressive. For example, on CUB, DSL outperforms domain-mix by +8.84% under 5-way 5-shot setting.
>
> Q3: The performance of domain-mix+teacher-regularization? Further, if we use the teacher network to calculate both BKLD loss and RCE loss for the domain-mix scheme, how about the performance?
>
> Ans: Thanks for the good question. Adding the teacher onto the domain-mix scheme only brings very slight improvement. On CUB (5-way 5-shot setting), the domain-mix learner achieves 64.73 accuracy, while "domain-mix+teacher" achieves 65.01 (+0.28%) accuracy. We infer it is because the domain-mix learner already well memorizes the domain-general knowledge as well as domain-specific knowledge (as analyzed in the Fig.1 caption in the manuscript), and thus the domain-general teacher does not provide additional domain-general knowledge. Another hint from this observation is that the domain-switch scheme is critical to our method.
>
> As for the question of "both BKLD loss and RCE loss", we are not quite clear on how to add the RCE loss into the domain-mix scheme. It is because under the domain-mix scheme, there is no domain-specific prompter, which is the prerequisite for the RCE loss. If we directly use the most adjacent model as the prompter, the resulting accuracy of "BKLD+RCE" shows no observable improvement (from 65.01 to 65.12). We are not sure whether we have configured the setup of this experiment exactly the same as you expected. Please let us know if you have further suggestions.
>
> [a] Haoqing Wang and Zhi-Hong Deng. Cross-domain few-shot classification via adversarial task augmentation. arXiv preprint arXiv:2104.14385, 2021.
>
> [b] Yanbin Liu, Juho Lee, Minseop Park, Saehoon Kim, Eunho Yang, Sung Ju Hwang, and Yi Yang. Learning to propagate labels: Transductive propagation network for few-shot learning. In 7th International Conference on Learning Representations, ICLR 2019.
>
> [c] Chen, W.Y., Liu, Y.C., Kira, Z., Wang, Y.C.F., Huang, J.B.: A closer look at few-shot classification.ICLR 2019.
>
> [d] Yunhui Guo, Noel C Codella, Leonid Karlinsky, James V Codella, John R Smith, Kate Saenko, Tajana Rosing, and Rogerio Feris. A broader study of crossdomain few-shot learning. ECCV 2020.

---

### Author Response · Authors · 2021-11-21
**Summary of revisions**

We thank all the reviewers for their valuable comments. We have uploaded a revised draft according to the helpful suggestions. We summarize the main changes as below:

1. We clarify in the introduction that "DSL favors domain-general knowledge over domain-specific knowledge" does not mean that DSL learns more domain general knowledge than the domain-mix learning scheme does (Page 2).

2. We clarify the caption of Table.1 and Table.2.

3. We add another dataset (Cars) into Table 3 to better compare the single-domain learner, domain-mix learner and the domain-switch learner. We show that the improvement of domain-mix learner is trivial, compared with the improvement of DSL.

4. We add the comparison of DSL against two transductive methods (TPN, GNN) into the appendix (A.4). For fair comparison, we use a simple label propagation method to cooperate with the proposed DSL.

---

### Author Response · Authors · 2021-11-30
**Thanks to all the Reviewers**

Dear reviewers,

We thank all the reviewers again for your professional reviews and valuable suggestions. We also appreciate Reviewer KRoA considers our reponses informative and raises his / her rating to "Accept".

We have done our best to address the questions raised in your review. The discussion period is coming to a close soon and we are still open to discussing any remaining concerns you may have until the very end. Please do not hesitate to raise your questions / concerns if there are any.

Thank you and best wishes.

Authors

---

### Decision · Program_Chairs · 2022-01-20

**Decision:**

Accept (Poster)

**Comment:**

This paper proposes an approach to improve cross-domain generalization in few-shot learning, using an objective that attempts to fight overfitting on the observed domain at any given iteration while maintaining the general learned information so far from all domains. The approach uses a domain-cycling procedure, where each iteration sees a single domain and, pseudo-labels coming from predictions of a previous iterate of the model and from a parameter-averaged general model are used in a combined training objective.

Three of the reviewers support acceptance (one strongly), while the fourth leans weakly towards rejection, despite an extensive response from the authors that include new results. One concern was a lack of comparison on Meta-Dataset, which the authors went some way towards addressing during the rebuttal, though they also argued Meta-Dataset couldn't really support the kind of cross-domain evaluation they were targeting. The reviewer was not convinced by the authors' argument, and I too am not, in particular when you consider that Meta-Dataset evaluations now often include evaluations on MNIST and CIFAR-10/100, in addition to MS-COCO and TrafficSigns (all not included in the training split of Meta-Dataset). That said, the experimental protocol favored in the authors' experiments certainly is sound and challenging for cross-domain generalization, so I'd hesitate to penalize them for that alternative choice.

Overall, I find the ideas behind this work neat, interesting and well motivated. Even if the basic ideas aren't completely novel, I found their combination thought provoking and creative.

Therefore, in the end, I feel this work will be beneficial to the body of literature on few-shot learning and would merit to appear at ICLR.